# Acute Myeloid Leukaemia and Acute Lymphoblastic Leukaemia Classification and Metabolic Characteristics for Informing and Advancing Treatment

**DOI:** 10.3390/cancers16244136

**Published:** 2024-12-11

**Authors:** Carrie Wemyss, Emily Jones, Régis Stentz, Simon R. Carding

**Affiliations:** 1Food, Microbiome and Health Research Programme, Quadram Institute Bioscience, Norwich NR4 7UQ, UK; c.wemyss@uea.ac.uk (C.W.); emily.jones@quadram.ac.uk (E.J.); regis.stentz@quadram.ac.uk (R.S.); 2Norwich Medical School, University of East Anglia, Norwich NR4 7TJ, UK

**Keywords:** acute myeloid leukaemia (AML), acute lymphoblastic leukaemia (ALL), risk stratification, metabolic vulnerabilities, targeted therapies, asparaginase, personalised medicine, drug resistance, metabolic heterogeneity, leukaemia stem cells

## Abstract

This review focuses on the classification, risk stratification, and metabolic characteristics of acute myeloid leukaemia (AML) and acute lymphoblastic leukaemia (ALL) and their importance in the advancement of treatment. It discusses current management strategies, including standard chemotherapy regimens and targeted therapies, while highlighting the challenges of drug toxicity and resistance. It also emphasises the importance of understanding metabolic vulnerabilities in leukaemia cells for developing more effective and personalised treatments. Special attention is given to asparaginase therapy and its mechanism of action, limitations, and potential for improvement in both ALL and AML treatment.

## 1. Introduction

Acute leukaemia, particularly acute myeloid leukaemia (AML) and acute lymphoblastic leukaemia (ALL), pose significant challenges in the field of haematological malignancies. Acute leukaemia impacts a diverse range of age groups. AML is more prevalent in adults, with 3100 new cases diagnosed in the UK each year. Despite treatment advances, the five-year survival rate of AML is still only 15%, with approximately 2600 deaths in the UK annually [1]. ALL is the most common childhood cancer, peaking in children aged 3–4 years, with 790 new cases and 230 deaths annually in the UK [1].

This review provides a comprehensive overview of the cell lineage and molecular and metabolic characteristics of these aggressive malignancies. This underpins the heterogeneity of the disease and the requirement for treatment to be personalised and tailored to individual cases. Understanding the specific cellular dependencies and vulnerabilities is crucial for future clinical advancements [2,3,4,5,6,7,8,9,10]. Treatment of acute leukaemia involves intensive chemotherapy regimens and targeted therapies, associated with significant toxicities [11,12]. These adverse effects range from myelosuppression and gastrointestinal issues to cardiotoxicity, pancreatitis, and secondary malignancies, necessitating a need for less toxic and more targeted treatments [13]. L-asparaginase has been used as a key component of ALL treatment protocols for the past 50 years and is particularly effective in lymphoblastic leukaemia due to these cells typically lacking the enzyme asparagine synthetase (ASNS) [14]. Exploration of the dual action of asparaginase against both asparagine and glutamine highlights its potential utility across different leukaemia subtypes. Studies in metabolic characterisation reveal potential pathways for further individualised use of asparaginase as perhaps a more targeted treatment than currently considered.

The integration of molecular and metabolic profiling in personalised medicine is central to developing effective treatment strategies tailored to individual patients’ needs [15] that minimise toxicity and maximise efficacy. While challenges remain in the fight against acute leukaemia, ongoing research aimed at the comprehensive classification and metabolic profiling of ALL and AML is a promising approach for enhancing treatment efficacy and improving outcomes for patients.

## 2. Pathophysiology

AML is a rapidly progressing heterogenous myeloid neoplasm characterised by the clonal expansion of myeloid progenitor cells in the bone marrow and peripheral blood. This proliferation primarily stems from the accumulation of diverse genomic and cytogenetic abnormalities, resulting in ineffective erythropoiesis, megakaryopoiesis, organ infiltration and bone marrow failure, causing the inadequate production of red blood cells and platelets [16].

ALL is an aggressive malignancy of B or T lymphoblasts, characterised by the uncontrolled proliferation of abnormal, immature lymphocytes and their progenitors. Environmental damage to DNA and genetic predisposition are among aetiologies that precede disease, causing lymphoid cells to undergo uncontrolled growth and spread throughout the body [17,18,19,20]. Like AML, this ultimately leads to the replacement of functional bone marrow elements and other lymphoid organs, leading to bone marrow failure. Furthermore, splenomegaly and hepatomegaly occur due to the sequestration of platelets and lymphocytes [21].

## 3. Classification and Risk Stratification, Informing Treatment and Prognosis

The World Health Organisation (WHO) classifies AML and ALL according to cell lineage, gene or chromosome changes, and cell differentiation (see Table 1). Acute leukaemia of ambiguous lineage (ALAL) and mixed-phenotype acute leukaemia (MPAL) have overlapping clinical and immunophenotypic features and are therefore grouped together. These have been found to share common molecular pathogenic mechanisms [22,23,24]. The separation of ALAL/MPAL allows for molecular classification, distinguishing those with genetic abnormalities from those defined by immunophenotyping. Cells may exhibit a pattern in which some leukaemia cells have myeloid features and others have lymphoid features, while some cells simultaneously display both myeloid and lymphoid features [25].

### 3.1. AML

AML is a heterogeneous disease requiring individualised cytogenetic and molecular characterisation (Table 1). Prognostic factors are subdivided into those related to either the patient or to the disease (Figure 1). The European LeukemiaNET (ELN) guidelines (2022) emphasise molecular characterisation and risk stratification as critical for prognostic classification and treatment strategy, categorising AML into favourable-, intermediate-, or adverse-risk groups (see Table 2) [28]. Despite advancements in therapeutic approaches, prognosis remains suboptimal, especially among older populations. Characterising fitness in the adult population is important when deciding treatment strategy.

### 3.2. ALL

Clinical factors and cytogenetic changes play an important role in risk stratification, which guides initial treatment regimen and allogeneic stem cell transplantation (Allo-SCT) (Figure 1 and Table 1). The Philadelphia chromosome, the t(9;22) cytogenetic aberration, has the greatest impact on prognosis and treatment. Prevalence in adult ALL can range from 15 to 30% and increases with age [30]. Response to initial therapy also predicts outcome. Patients are evaluated for minimal residual disease (MRD) using molecular techniques such as flow cytometry and PCR [31]. Bruggemann et al. re-stratified standard-risk patients to low risk (<10^−4^), intermediate risk, and high risk (>10^−4^), with relapse rates of 0%, 47%, and 94%, respectively, based on the persistence of elevated MRD, defined as >10^−4^ [32]. The National Comprehensive Cancer Network (NCCN) has developed recommendations to approach risk stratification [29]. The NCCN recognises that adolescent and young adults (AYAs) (those aged 15–39 years) may benefit from treatment with paediatric-inspired regimens, with adults (>40 years) considered separately [33]. Both age groups are then stratified into subgroups: high-risk Ph-positive (Philadelphia chromosome-positive ALL) and standard-risk Ph-negative. The Ph-negative subgroup can further be categorised as high, intermediate, or low risk (Table 3), with the 5-year overall survival rates based on risk categories being 5%, 34%, and 55% respectively [34].

## 4. Treating Acute Leukaemia

### 4.1. Drug Classification

Cancer chemotherapy drugs are divided into two main categories: non-targeted agents with broad specificity, and targeted drugs developed for specific molecular targets on cancer cells.

### 4.2. Current Standard of Care

Treatment of ALL and AML typically follows an induction, maintenance, and consolidation regime. The induction phase typically involves intensive chemotherapy combination regimens and aims to achieve complete remission (Table 4 and Table 5). Targeted therapies are incorporated into induction protocols to improve outcomes in specific genetic and molecular subgroups (Table 6). Newly developed targeted therapies include small-molecule inhibitors such as those that target FLT3, IDH, and BCL-2, as well as immunotherapies including Chimeric Antigen Receptor (CAR) T-cell therapy and bispecific T-cell engagers (BiTEs). Additionally, there are antibody–drug conjugates and emerging Menin inhibitors which are currently under investigation for use in KMT2A rearrangements or NPM1 mutations in both ALL and AML. The optimal sequence and combination of these targeted agents with standard chemotherapy is an active area of research in AML [35,36]. Less intensive options are considered for older adults or those with significant comorbidities who cannot tolerate intensive chemotherapy.

### 4.3. Adverse Reactions

Non-targeted anti-leukemic drugs are cytotoxic; suppress haematopoiesis; and can cause cutaneous eruptions, vascular damage, and lung and liver injury (Table 5). Current research into dose optimisation aims to adjust drug doses and schedules to maintain efficacy while minimising adverse effects [75]. Further attempts to decrease the overall toxicity of leukaemia treatments focuses on increasing the specificity of drugs. Targeting molecular pathways specific to leukaemia cells has potential to spare healthy cells and reduce systemic adverse effects by reducing the dose of traditional chemotherapy agents required [12]. Adverse effect profiles of targeted treatments tend to be more manageable; however, these include lengthened QT intervals, a conduction disorder of the heart, febrile neutropenia, cytopenia, infections, gastrointestinal issues, and skin disorders (Table 6). Additionally, targeted immune activation aims to harness the immune system to deliver precise treatment with reduced off-target effects [76].

### 4.4. Hypersensitivity Reactions

In chemotherapy treatment for acute leukaemia, various adverse effects may have an immune basis, including drug-induced thrombocytopenia, neutropenia, and anaemia; vascular disorders; liver injury; lung disease; and various dermatological manifestations (Table 5). Certain drugs cause specific hypersensitivity reactions, such as cytarabine, causing a Type IV, delayed, and T cell-mediated reaction and L-asparaginase, causing a Type I, IgE antibody-mediated reaction (Table 5 and Table 7).

## 5. Metabolic Characteristics, Vulnerabilities, and Treatment Strategy

Metabolic vulnerabilities in leukaemia cells present significant opportunities for targeted therapies by exploiting the unique dependencies of these cells. Understanding and targeting the metabolic vulnerabilities of leukaemia cells offers a promising strategy for developing more effective and personalised treatments for leukaemia.

### 5.1. Key Differences in Metabolic Profiles Between Lymphoid and Myeloid Leukaemia Cells

#### 5.1.1. Glycolysis and Oxidative Phosphorylation

Lymphoid leukaemia cells rely more on oxidative phosphorylation (OxPhos) rather than glycolysis, exhibiting a reduced glucose uptake in comparison to normal hematopoietic stem cells (HSCs), with increased mitochondrial respiration and reactive oxygen species (ROS) production [2,3]. Myeloid leukaemia cells show significant alterations in glycolytic pathways, characterised by a different set of metabolic adaptations that may include both glycolysis and OxPhos, depending on the specific subtype of myeloid leukaemia [4]. AML cells can alter the expression of glycolytic enzymes, upregulating glycolysis and switching between glycolysis and OxPhos depending on environmental conditions [85,86].

#### 5.1.2. Amino Acid Metabolism

In lymphoid leukaemia cells, there is protective activity in glutathione metabolism, involving the overexpression of enzymes like glutamine dehydrogenase, which is crucial for glutathione synthesis [5,6,7]. Myeloid leukaemia cells also exhibit the aberrant regulation of glutathione, with significantly lower levels of reduced glutathione (GSH), higher levels of oxidised glutathione (GSSG) and reduced total glutathione. In addition, studies observe a decreased GSH-to-GSSG ratio in CD34^+^ AML cells [8].

#### 5.1.3. Lipid Metabolism

Lymphoid leukaemia cells demonstrate active lipid metabolism, with an accumulation of ceramide and lipoprotein lipases, making them susceptible to fatty acid oxidation (FAO) inhibitors. This is observed to be useful in cases of treatment resistance [9]. Myeloid leukaemia cells undergo the metabolic reprogramming of lipid metabolism crucial for tumorigenesis and disease progression, supporting processes such as invasion, metastasis, and abnormal signalling [10]. Pathway alterations and their implications can differ based on genetic and environmental factors. Leukaemia cells treated with L-asparaginase have been shown to react by reprogramming their metabolism to increase fatty acid oxidation (FAO) and autophagy to compensate for asparagine and glutamine depletion [87]. The pharmacological inhibition of FAO increases sensitivity to asparaginase in leukaemia cells, supporting the theory of their pro-survival effect and potential role in the mechanism of treatment resistance [87].

### 5.2. Key Insights into How These Vulnerabilities Are Being Utilised for Therapeutic Purposes

#### 5.2.1. Targeting Specific Metabolic Pathways

Leukaemia cells exhibit altered metabolic pathways that support their rapid proliferation and survival. These include glycolysis, the oxidation of fatty acids, and the Krebs cycle. Targeting these pathways can disrupt the energy supply and biosynthetic processes crucial for leukaemia cell survival.

#### 5.2.2. Leukaemia Stem Cells

Leukaemia stem cells (LSCs) resistant to conventional treatments have distinct metabolic preferences, such as heavy reliance on OXPHOS and specific amino acid metabolisms [88]. Inhibiting these pathways, for example, by targeting the enzyme nicotinamide phosphribosyltransferase (NAMPT), can reduce OXPHOS, with potential to eradicate resistant LSCs, while sparing normal cells [89].

#### 5.2.3. Combination Therapies

Combining metabolic inhibitors with chemotherapy or immunotherapy can enhance therapeutic efficacy. Venetoclax combined with azacytidine disrupts energy metabolism in AML cells, targeting both blasts and LSCs, and restores sensitivity to treatment in relapsed or refractory cases of AML [90]. Additionally, 2-deoxy-D-glucose (2-DG) interferes with D-glucose metabolism, enhancing the anti-cancer effects of idarubicin (IDA) in IDA-resistant P388 leukaemia cells [91].

#### 5.2.4. Personalised Medicine

Leukaemia cells display distinct metabolic states and adaptation mechanisms, serving as potential targets for treatment. However, the heterogeneity of the disease advocates for therapies to be tailored more individually. AML displays significant metabolic heterogeneity between patients within genetically distinct subclones [92,93,94]. The serine/threonine protein kinase PDK1 acts as a targetable determinant of different metabolic states in AML, functioning as a gatekeeper of glycolysis by phosphorylating and inactivating pyruvate dehydrogenase (PDH) [95]. Studies identified two main metabolic states in AML: PDK1-low, which is OXPHOS-driven, and PDK1-high, which is associated with low OXPHOS and an increase in stemness transcriptional signatures [85]. Insights into metabolic differences alongside clonal heterogeneity in AML allow for personalised, subclone-specific targeting strategies.

In addition to metabolic heterogeneity, the metabolic plasticity of leukaemia cells also presents a challenge in treatment design. With flexibility to undergo compensatory metabolic and energetic adaptations in response to the inhibition of metabolic pathways, leukaemia cells can adapt and survive when specific metabolic pathways are targeted therapeutically [77,96]. LSC plasticity in mixed-lineage leukaemia-rearranged B-lymphoblastic leukaemia (MLL-r B-ALL) allows the cells to switch lineages and emerge from differentiated populations, seen most frequently when under chemotherapy pressure [97]. AML stem cells can switch between a low-cycling chemotherapy-resistant state and an actively proliferating state [77]. This plasticity allows for interruption in drug efficacy, contributing to treatment resistance and disease progression.

## 6. Asparaginase as Targeted Treatment in Acute Leukaemia

In healthy cells, the non-essential amino acid, asparagine, can be synthesised by the enzymatic action of asparagine synthetase (ASNS) or obtained from the diet. Sufficient levels of cellular asparagine are required for DNA, RNA, and protein synthesis. Depleted asparagine levels ultimately lead to the activation of apoptotic cell death mechanisms (Figure 2).

L-asparaginase has been a key component of ALL treatment protocols for the past fifty years and has been found to be particularly effective in lymphoblastic leukaemia due to cells typically lacking ASNS [14]. Lymphoblastic leukaemia cells are also naturally susceptible to asparagine depletion. Cells unable to produce asparagine on their own are heavily dependent on extracellular sources. At sufficient activity levels, asparaginase depletes serum L-asparagine, eventually leading to leukaemic cell death.

### 6.1. Current Limitations of Asparaginase Therapy

Despite their effectiveness, the high toxicity and side effects of current L-asparaginase formulations limit optimal clinical application, with lack of tolerance often leading to treatment interruption and discontinuation (Table 7). Current studies aim to improve treatment adherence, to reduce treatment interruption, and to improve the quality of life of patients receiving this drug. Combining L-asparaginase with other drugs, such as FAOs, may improve efficacy while reducing adverse effects. The development of L-asparaginase variants with reduced L-glutaminase coactivity has been found to reduce acute toxicity compared to FDA-approved high L-glutaminase enzymes, while maintaining anti-leukaemic efficacy [98]. The short half-lives of current L-asparaginase formulations have prompted efforts to develop variants with longer half-lives, which correlate with lower immunogenicity and reduced toxicity, while maintaining efficacy [99]. Paediatric studies emphasise the importance of monitoring in the early detection of asparaginase-related toxicity, in addition to understanding risk factors for toxicity that can inform individualised treatment approaches [100]. Identified risk factors include the presence of the TEL-AML1 fusion gene, which is associated with a higher risk of clinical hypersensitivity. Additionally, T-cell ALL correlates with a higher risk of hepatotoxicity in comparison to B-cell ALL. Other identified factors include age, gender, response to treatment, and presence of central nervous system (CNS) infiltration at diagnosis.

### 6.2. Obstacles to Asparaginase Use in AML

AML cells are, on average, 7-fold more resistant to L-asparaginase than ALL cells [101]. This may be due to AML cells having variable expression of ASNS, in comparison to ALL cells. However, ASNS activity in blast cells of acute monocytic leukaemia (10% of AML cases) are reported to have the lowest median of ASNS activity, with activity varying over only a fourfold range [14]; therefore, certain subgroups of AML may show more promising outcomes when treated with L-asparaginase. Lysosomal cysteine protease cathepsin B (CTSB), asparaginyl endopeptidase, and neutralising antibodies may also play a role in resistance and sensitivity to L-asparaginase [102,103]. L-asparaginase should therefore be tested in combination with other drugs that counteract factors decreasing its efficacy, such as the inhibition of ASNS and/or CTSB with specific protease inhibitors. With its potential to downregulate transcription of the ASNS gene, cytarabine shows promising synergetic effects with L-asparaginase. For example, asparaginase intensification studies, consisting of a combination of high doses of cytarabine and asparaginase (with noted importance of the drug sequence, with asparaginase following cytarabine) was found to eliminate the benefit of prolonged maintenance therapy in childhood AML. This was accompanied by an overall improvement in survival and improved complete remission rate [104]. The efficacy of asparaginase was also observed when combined with methotrexate in both adult and paediatric refractory or relapsed AML [105,106]. L-asparaginase may have anti-leukaemic activity in AML cells in general, and leukaemic stem cells in particular, if the protective effect of the bone marrow microenvironment can be overcome. A proposed approach to counteract this protective effect on residual AML cells, is to interfere with the binding of residual cells to the bone marrow stroma with CXCR4 chemokine-receptor antagonists, such as plerixafor [107].

Although AML cells may have variable expression of ASNS, they are particularly susceptible to glutamine depletion [103]. While L-asparaginase’s primary target is asparagine, it has significant activity against glutamine (Figure 2). L-asparaginase’s primary activity hydrolyses L-asparagine into L-aspartic acid and ammonia, with secondary hydrolysis of L-glutamine into L-glutamic acid and ammonia (glutaminase activity). The glutaminase activity of FDA-approved L-asparaginases ranges from 2% to 10% of their primary asparaginase activity [98]. AML cells and LSCs rely heavily on glutamine metabolism for critical cellular processes and show increased sensitivity to glutamine depletion compared to normal cells [108]. Additionally, the genetic knockdown or pharmacologic inhibition of glutaminase negatively affects AML cells, while sparing normal CD34^+^ HSCs [109]. The glutaminase activity of L-asparaginase enhances the drug’s anticancer effects in ASNS-positive cancer cells [98]. Further studies have also found that the glutaminase activity of L-asparaginase was necessary for durable anticancer activity in vivo, even against ASNS-negative cancer types [110].

### 6.3. Metabolic Characteristics and Sensitivity to Asparaginase Treatment

Leukaemia cells cluster into distinct groups based on their metabolic profiles, which correlate with their response to asparaginase treatment. In a recent study describing the relationship between the metabolic profile of leukaemia cells and their sensitivity to commonly used anti-leukaemic drugs, cells sensitive to asparaginase were ranked in a lower glycolytic cluster [15]. Indeed, a significant correlation was observed between higher ATP-linked respiration, lower basal mitochondrial membrane potential, and increased sensitivity to asparaginase. No similar correlation was found for other cytostatic drugs tested [15]. The findings suggest that the metabolic profile of leukaemia cells plays a crucial role in determining the efficacy of asparaginase treatment, and that metabolic profiling could be used to predict asparaginase sensitivity in leukaemia patients. Thus, patients with higher glycolytic activity might benefit from alternative or additional treatments to asparaginase. Targeting specific metabolic pathways (e.g., glycolysis or mitochondrial respiration) could potentially enhance the efficacy of asparaginase treatment, especially in less sensitive patients who would otherwise be more likely to fail the conventional therapy without having any other detectable risk factors. Characterising leukaemic cell and blast metabolic state, at the time of diagnosis, may help identify patients who are particularly sensitive, or with lower sensitivity to asparaginase, and offer a potential pathway for personalised treatment strategies in leukaemia therapy.

However, the integration of metabolic profiling into routine clinical practice is hindered by several complex practical challenges. The techniques used to evaluate cellular metabolism include sophisticated stable isotope tracing and biochemical assays, which can be difficult to standardise and are not ideally suited to high-throughput clinical and diagnostic labs and are not currently used within UK National Health Services. More recent methods like extracellular flux analysis using Seahorse Analysers offer high throughput but require meticulous optimisation and set up which is why they are more commonly found in research settings. The cost of these technologies may also be a prohibiting factor for clinical laboratories along with the requirement for highly trained staff. Also, there is no single gold-standard test for metabolic diagnostics as metabolic activity often involves a sequence of events along a gradient of pathway flux. Consequently, it is likely that multiple tests will be necessary to accurately profile metabolic characteristics, posing challenges in terms of costs, accessibility, and standardisation.

## 7. Conclusions

The treatment landscape for acute leukaemia is evolving due to advances in our understanding of their cellular and metabolic characteristics, and the targeting of these for therapeutic benefit. Statistics surrounding these diseases highlight the urgency for improved treatment options. With AML resulting in over 2600 deaths each year and a mere 15% five-year survival rate despite treatment advances, and ALL being the most common childhood cancer, it is evident that more effective and better tolerated therapies are critically needed.

The toxicity profile of current leukaemia treatments remains a significant challenge, impacting patient quality of life and limiting therapeutic options; however, the exploration of metabolic characteristics reveals potential pathways for targeted therapies. The effectiveness of L-asparaginase in treating ALL exemplifies how understanding metabolic dependencies can inform treatment strategies. Individual susceptibility to asparaginase therapy has potential to inform its utilisation across the subtypes of leukaemia on a case-by-case basis, perhaps changing the outlook on asparaginase as an increasingly targeted therapy for both ALL and AML. This prompts the question of whether additional screening prior to therapy is necessary, which highlights the need to better understand the practical limitations of integrating these investigations into clinical practice, and may also involve considering the potential outsourcing of clinical investigations.

Continued collaboration among researchers, clinicians, and patients will be essential to navigate these scientific insights into clinical practice. While challenges remain in the fight against acute leukaemia, ongoing research into more personalised medicine offers promising avenues for enhancing treatment efficacy, while reducing toxicity, and provides hope for improved outcomes for all patients affected by AML and ALL.

## Figures and Tables

**Figure 1 cancers-16-04136-f001:**
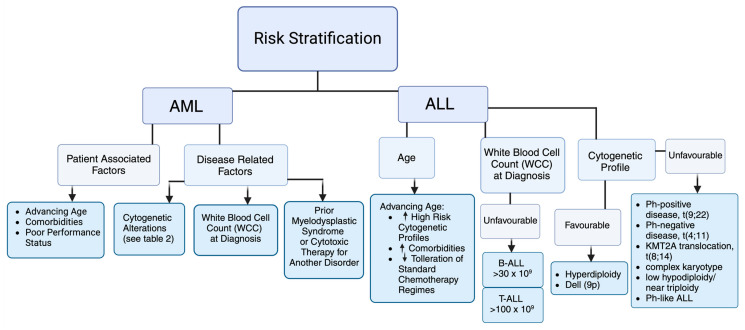
Risk stratification in AML and ALL as per European LeukemiaNET (ELN) guidelines and the National Comprehensive Cancer Network (NCCN) [28,29].

**Figure 2 cancers-16-04136-f002:**
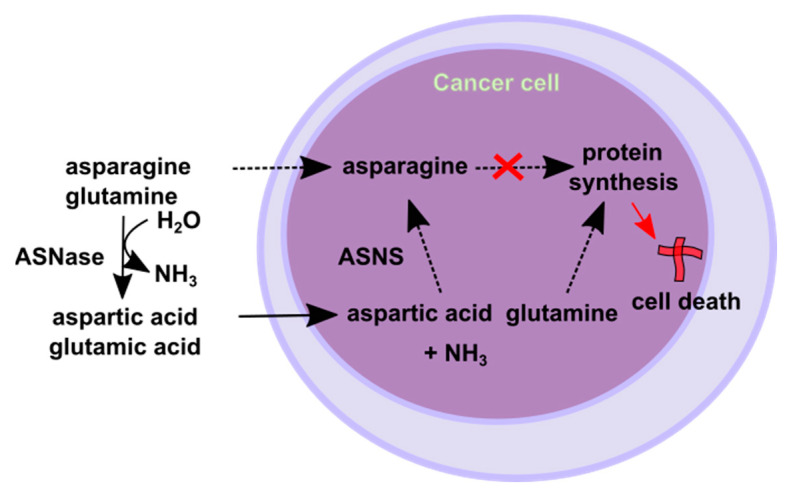
Mechanism of action of asparaginase in the treatment of leukaemia. ASNase: L-asparaginase; ASNS: L-asparagine synthetase; dashed arrows: reduced or missing.

**Table 1 cancers-16-04136-t001:** WHO classification of AML and ALL.

Cell Lineage	Classification	Subtype
Lymphoid	B-cell ALL with certain genetic abnormalities	B-cell ALL with hypodiploidy B-cell ALL with hyperdiploidy B-cell ALL with t(9;22) (Philadelphia chromosome, BCR-ABL1 fusion) B-cell ALL with translocation involving chromosome 11 B-cell ALL with t(12;21) B-cell ALL with t(1;19) B-cell ALL with t(5;14) B-cell ALL with iAMP21 * B-cell ALL with BCR-ABL1–like ALL * B-cell ALL, not otherwise specified
	T-cell ALL	Early T-cell precursor lymphoblastic leukaemia *
Myeloid	AML with defining genetic abnormalities	Acute promyelocytic leukaemia with PML::RARA fusion AML with RUNX1::RUNX1T1 fusion AML with CBFB::MYH11 fusion AML with DEK::NUP214 fusion AML with RBM15::MRTFA fusion AML with BCR::ABL1 fusion AML with KMT2A rearrangement AML with MECOM rearrangement AML with NUP98 rearrangement AML with NPM1 mutation AML with CEBPA mutation AML, myelodysplasia-related AML with other defined genetic alterations
	AML, defined by differentiation	AML with minimal differentiation AML without maturation AML with maturation Acute basophilic leukaemia Acute myelomonocytic leukaemia Acute monocytic leukaemia Acute erythroid leukaemia Acute megakaryoblastic leukaemia
Acute leukaemia of ambiguous lineage (ALAL) and mixed-phenotype acute leukaemia (MPAL)	ALAL/MPAL with defining genetic abnormalities	Mixed-phenotype acute leukaemia with BCR::ABL1 fusion Mixed-phenotype acute leukaemia with KMT2A rearrangement Acute leukaemia of ambiguous lineage with other defined genetic alterations Mixed-phenotype acute leukaemia with ZNF384 rearrangement Acute leukaemia of ambiguous lineage with BCL11B rearrangement
	ALAL, immunophenotypically defined	Mixed-phenotype acute leukaemia, B/myeloid Mixed-phenotype acute leukaemia, T/myeloid Mixed-phenotype acute leukaemia, rare types Acute leukaemia of ambiguous lineage, not otherwise specified Acute undifferentiated leukaemia

* (from [26,27]): * Provisional entity.

**Table 2 cancers-16-04136-t002:** ELN-recommended stratification of molecular and cytogenic alterations in AML.

Risk Profile	Subsets
Favourable	t(8;21)(q22;q22); RUNX1-RUNX1T1 inv(16)(p13.1;q22); or t(16;16)(p13.1;q22); CBFB-MYH11 Mutated NPM1 without FLT3-ITD (normal karyotype) Biallelic mutated CEBPA (normal karyotype)
Intermediate-I	Mutated NPM1 and FLT3-ITD (normal karyotype) Wild-type NPM1 and FLT3-ITD (normal karyotype) Wild-type NPM1 without FLT3-ITD (normal karyotype)
Intermediate-II	t(9;11)(p22;q23); MLLT3-KMT2A Cytogenic abnormalities not classified as favourable or adverse
Adverse	inv(3)(q21;q26.2) or t(3;3)(q21;q26.2); GATA2-MECOM (EVI1) t(6;9)(p23;q34); DEK-NUP214 t(v;11)(v;q23); KMT2A rearranged −5 or del(5q); −7; abnl(17p); complex karyotype

From [28].

**Table 3 cancers-16-04136-t003:** Standard-risk pH-negative acute lymphoid leukaemia (ALL) subdivision.

Risk Profile	Cytogenetics	Other Factors	MRD
High	Unfavourable cytogenetics ^a^	OR Age > 35 AND Elevated white blood cell (WBC) count ^b^	OR MRD (>10^−4^)
Intermediate	No risk factors based on cytogenetics	Age > 35 OR Elevated WBC count	MRD (<10^−4^)
Low	No risk factors based on cytogenetics	No risk factors based on: Age OR WBC count	MRD (<10^−4^)

From [29]. ^a^ Unfavourable cytogenetics defined in Figure 1. ^b^ Elevated WBC count (>30 × 10^9^ for B-ALL or >100 × 10^9^ for T-ALL).

**Table 4 cancers-16-04136-t004:** Standard induction regimes for AML and ALL.

Disease	Standard Induction Treatment	Targeted Drugs	Additional Drugs
AML	“7+3” regimen: Cytarabine (7-days) AND Anthracycline (3 days) - Daunorubicin - Idarubicin	For FLT3-mutated AML: - Midostaurin For CD33-positive AML: - Gemtuzumab ozogamicin For therapy-related AML: - Liposomal daunorubicin-cytarabine For IDH1/2-mutated AML: - Ivosidenib - Enasidenib	Less intensive options: - Azacitidine, decitabine - Low-dose cytarabine - Venetoclax, combined with low-dose cytarabine or hypomethylating agents - Glasdegib plus low-dose cytarabine - Ivosidenib +/− azacitidine - Enasidenib
ALL	Vincristine AND Anthracycline - Daunorubicin - Doxorubicin AND A steroid - Dexamethasone - Prednisone	For Ph+ ALL: - Imatinib - Dasatinib	As per risk stratification: - Cyclophosphamide - L-asparaginase - Pegaspargase - Crisantaspase recombinant - High-dose methotrexate - High-dose cytarabine

**Table 5 cancers-16-04136-t005:** Non-targeted chemotherapy drugs used in the treatment of AML and ALL.

Drug	Type	Indication	Common Adverse Effects	Serious Adverse Effects	Reference
Cytarabine	Antimetabolite	AML, ALL	Myelosuppression, nausea/vomiting, diarrhoea, mucositis	Cerebellar toxicity, acute pulmonary syndrome, hepatotoxicity	[37]
Daunorubicin	Anthracycline	AML, ALL	Myelosuppression, cardiotoxicity, alopecia, mucositis	Cardiotoxicity, secondary malignancies	[38,39]
Idarubicin	Anthracycline	AML	Myelosuppression, cardiotoxicity, alopecia, mucositis	Cardiotoxicity, hepatotoxicity	[40]
Mitoxantrone	Anthracenedione	AML	Myelosuppression, cardiotoxicity, alopecia	Cardiotoxicity, therapy-related acute myeloid leukaemia	[41]
Etoposide	Topoisomerase II inhibitor	AML	Myelosuppression, nausea/vomiting, alopecia	Secondary leukaemias, anaphylaxis	[42,43]
Methotrexate	Antimetabolite	ALL	Myelosuppression, mucositis, hepatotoxicity, nephrotoxicity	Neurotoxicity, severe mucositis, acute kidney injury, hepatotoxicity	[44,45]
Vincristine	Vinca alkaloid	ALL	Peripheral neuropathy, constipation	Severe neurotoxicity, paralytic ileus	[46,47]
L-Asparaginase	Enzyme	ALL	Hypersensitivity reactions, pancreatitis, coagulopathy	Pancreatitis, thrombosis	[48,49,50,51]
6-Mercaptopurine	Antimetabolite	ALL	Myelosuppression, hepatotoxicity	Severe hepatotoxicity, pancreatitis	[52]
Cyclophosphamide	Alkylating agent	ALL	Myelosuppression, haemorrhagic cystitis, alopecia	Haemorrhagic cystitis, cardiotoxicity, secondary malignancies	[13,53,54]
Azacitidine	Hypomethylating agent	AML (older/unfit patients)	Myelosuppression, nausea/vomiting, injection site reactions	Tumour lysis syndrome, renal failure	[55,56]
Decitabine	Hypomethylating agent	AML (older/unfit patients)	Myelosuppression, fatigue, nausea	Severe infections	[57]

From [58,59].

**Table 6 cancers-16-04136-t006:** Targeted chemotherapy drugs used in the treatment of AML and ALL.

Drug	Target	Type	Indication	Common Adverse Effects	Serious Adverse Effects	Reference
Venetoclax	BCL2	BCL-2 Inhibitor	AML	Nausea, diarrhoea, fatigue	Tumour lysis syndrome, severe myelosuppression	[60]
Midostaurin	FLT3	FLT3 Inhibitor	FLT3-mutated AML	Nausea, vomiting, headache	QT prolongation, interstitial lung disease	[12]
Gilteritinib	FLT3	FLT3 Inhibitor	FLT3-mutated AML	Myalgia, transaminase elevation	Differentiation syndrome, posterior reversible encephalopathy syndrome	[61]
Ivosidenib	IDH1	IDH Inhibitor	IDH1-mutated AML	Fatigue, nausea, diarrhoea	Differentiation syndrome, QT prolongation	[62]
Enasidenib	IDH2	IDH Inhibitor	IDH2-mutated AML	Nausea, diarrhoea, decreased appetite	Differentiation syndrome, liver toxicity	[63]
Gemtuzumab ozogamicin	CD33	Antibody–Drug Conjugate	CD33+ AML	Fever, nausea, infection	Veno-occlusive disease, severe myelosuppression	[64]
Glasdegib	Hedgehog pathway	Hedgehog Pathway Inhibitor	AML	Muscle spasms, alopecia, fatigue	QT prolongation, embryo/foetal toxicity	[65]
Imatinib	BCR-ABL	Tyrosine Kinase Inhibitor (TKI)	Ph+ ALL	Nausea, vomiting, diarrhoea, muscle cramps, fluid retention	Myelosuppression, hepatotoxicity	[66]
Dasatinib	BCR-ABL	Tyrosine Kinase Inhibitor (TKI)	Ph+ ALL	Diarrhoea, nausea, headache, muscle/joint pain, fluid retention	Myelosuppression, pleural effusion, pulmonary arterial hypertension, QT prolongation, pancreatitis	[67,68,69]
Blinatumomab	CD19, bispecific antibody	Bispecific T-cell Engager	Relapsed or refractory B-cell ALL	Fever, headache, nausea	Cytokine release syndrome, neurotoxicity	[70,71]
Inotuzumab ozogamicin	CD22, antibody–drug conjugate	Antibody–Drug Conjugate (ADC)	Relapsed or refractory B-cell ALL	Thrombocytopenia, neutropenia, infections	Veno-occlusive disease, increased risk of infections	[72]
Tisagenlecleucel	CD19	CART-cell	B-cell ALL, relapsed or refractory B-cell ALL	Fever, cytopaenia, headache, oedema, nausea, fatigue	Cytokine release syndrome (CRS), neurological toxicities, increased risk of infections, anaphylaxis, T-cell malignancies	[73]
brexucabtagene autoleucel	CD19	CART-cell	B-cell ALL, relapsed or refractory B-cell ALL	Fever, cytopaenia, headache, oedema, nausea, constipation, fatigue	Cytokine release syndrome (CRS), neurological toxicities, increased risk of infections, renal and respiratory problems	[74]

**Table 7 cancers-16-04136-t007:** Adverse effects of asparaginase therapy.

Body System	Common Adverse Effects	Serious Adverse Effects	Mechanism of Action of Adverse Effects	References
Gastrointestinal	- Nausea - Vomiting - Diarrhoea - Loss of appetite	- Pancreatitis	- Glutamine helps maintain the health of the intestinal mucosa. Depletion of glutamine contributes to gastrointestinal disturbances. - Interferes with the highly active pancreatic protein synthesis. - Upregulates asparagine synthetase (ASNS) expression in acinar cells. - Increases intracellular calcium levels in pancreatic acinar cells, leading to calcium overload, which can cause cell damage and necrosis. - Causes premature activation of trypsin within pancreatic acinar cells, acinar cell destruction, inflammation, and autodigestion.	[48,77,78]
Haematological	- Thrombocytopenia - Anaemia	- Thromboembolism	- Decreased synthesis of factors involved in coagulation and fibrinolysis due to reduced protein availability, increasing risk of thrombosis or bleeding. - Myelosuppression.	[49,50,51,79]
Neurological	- Fatigue - Headache	- Central nervous system toxicity—seizures, confusion	- Depletion of plasma proteins involved in coagulation and fibrinolysis leads to both thrombotic and haemorrhagic events in the brain. - Hyperammonaemia can lead to a diffuse encephalopathy. - Direct toxic effects causing reversible posterior leukoencephalopathy syndrome.	[80]
Dermatological	- Urticaria	- Severe allergic reactions	- Immune response to foreign proteins in asparaginase leads to hypersensitivity reactions.	[81]
Hepatic	- Elevated transaminases	- Hepatotoxicity	- Disruption of protein synthesis affecting liver function.	[82]
Metabolic	- Weight loss - Changes in taste	- Hyperglycaemia	- L-asparaginase hydrolyses asparagine, a key component of insulin. Depletion of asparagine leads to reduced insulin synthesis in pancreatic beta cells.	[83,84]
Renal	- Mild changes in kidney function	- Acute kidney injury	- Hyperammonaemia contributes to electrolyte disturbance and dehydration. - Reduced renal perfusion and function due to dehydration and thrombosis.	[49,50,51]
Cardiovascular	- Peripheral oedema	- Thrombotic events (e.g., stroke, myocardial infarction)	- Reduced plasma oncotic pressure due to reduced albumin levels, causing fluid to leak from the intravascular space into the interstitial space. - Coagulopathy due to decreased synthesis of fibrinogen and antithrombin III.	[49,50,51]

## Data Availability

No new data were created or analysed in this study. Data sharing is not applicable to this article.

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
