# Peer review of "Acute Myeloid Leukaemia and Acute Lymphoblastic Leukaemia Classification and Metabolic Characteristics for Informing and Advancing Treatment"

_cancers, 2024, doi:10.3390/cancers16244136_

Round 1
Reviewer 1 Report
Comments and Suggestions for Authors
Overall this is an informative and well written review that focuses on the importance of understanding metabolic characteristics of AML and ALL as a potential strategy for developing more effective and personalised treatments.
A novel gap in knowledge has been identified and addressed, and the references cited are appropriate and relevant, however, only 42 out of 110 references are from the last 5 years, and several are very old. This review would benefit from including more recent literature to replace some of the references that are 10 years or more older.
The statements and conclusion are coherent and supported by the provided citations. Some of the tables require changing, namely:
· Table 3 is missing column headings
· Table 4 – the information appears to be mixed up between AML and ALL (are the 2 rows labelled correctly?) – please also include column headings
Author Response
Response to Reviewer 1 comments
We thank the reviewer for their thoughtful comments and suggestions for improving our manuscript. We have below provided a point-by-point response to each comment/query.
A novel gap in knowledge has been identified and addressed, and the references cited are appropriate and relevant, however, only 42 out of 110 references are from the last 5 years, and several are very old. This review would benefit from including more recent literature to replace some of the references that are 10 years or more older.
We have updated the bibliography by replacing where appropriate older references with more recent ones. Please see references 17-20, 33, 34,108 and 109.
The statements and conclusion are coherent and supported by the provided citations. Some of the tables require changing, namely:
- Table 3 is missing column headings
This has been corrected.
- Table 4 – the information appears to be mixed up between AML and ALL (are the 2 rows labelled correctly?) – please also include column headings
This has been corrected.
Reviewer 2 Report
Comments and Suggestions for Authors
Wemyss et al. submitted an interesting article providing a comprehensive review of AML and ALL. The authors emphasized the importance of classification, risk stratification, and understanding metabolic vulnerabilities in leukemia cells. They focused on current treatment modalities and challenges like drug resistance, as well as the therapeutic potential of asparaginase in both AML and ALL. The review effectively integrates molecular and metabolic profiling to advocate for personalized treatment strategies aimed at enhancing patient outcomes. The manuscript is well-written and organized into several clear sections. However, I have some comments and suggestions that might further improve the manuscript prior to publication:
1. It would be beneficial for the authors to discuss in more detail the practical challenges of integrating metabolic profiling into routine clinical practice.
2. It is also suggested to include more information about other therapeutic drugs, such as immunotherapies or new small-molecule inhibitors.
3. It would be advantageous if the authors addressed practical limitations, such as the cost, accessibility, and standardization of metabolic profiling techniques.
4. Novel therapies such as CAR-T cells, antibody-drug conjugates, or bispecific antibodies should be included and discussed.
5. I like the conclusion, but I suggest adding more actionable recommendations for future research or clinical applications, in addition to the general advocacy for personalized medicine.
Author Response
Response to Reviewer 2 comments
We thank the reviewer for their thoughtful comments and suggestions for improving our manuscript. We have below provided a point-by-point response to each comment/query.
The manuscript is well-written and organized into several clear sections. However, I have some comments and suggestions that might further improve the manuscript prior to publication:
- It would be beneficial for the authors to discuss in more detail the practical challenges of integrating metabolic profiling into routine clinical practice.
This has been addressed in section 6.3., where we have discussed techniques used to assess or measure cellular metabolism, including sophisticated stable isotope tracing and biochemical assays. We have highlighted their current, mostly experimental, use in research laboratories, and the practical challenges in incorporating these into clinical and diagnostic laboratories, which include high costs, specialised and expensive technologies, difficulties in standardisation, and need for highly trained staff.
Furthermore, we have discussed that due to the nature of metabolic activity, there is possibility that there may be need for more than one test, without one gold standard test for metabolic diagnostics, which could be a contributing factor to high costs, and challenges in accessibility and standardisation.
- It is also suggested to include more information about other therapeutic drugs, such as immunotherapies or new small-molecule inhibitors.
This information has been included in section 4.2. and Table 6, where we refer to novel targeted agents including small-molecule inhibitors such as FLT3, IDH and BCL-2 inhibitors, immunotherapies such as Chimeric Antigen Receptor (CAR) T-cell therapy and Bispecific T-cell engagers (BiTEs), Antibody-Drug conjugates, and the emerging Menin inhibitors, currently under investigation for use in KMT2A rearrangements or NPM1 mutations in both ALL and AML. Drugs of these types that are currently used in clinical practice are listed in Table 6, including their target, indication for use, commonly adverse effects and serious adverse effects.
- It would be advantageous if the authors addressed practical limitations, such as the cost, accessibility, and standardization of metabolic profiling techniques.
We have included a brief discussion of techniques used to assess or measure cellular metabolism, including sophisticated stable isotope tracing and biochemical assays. We have discussed their current, mostly experimental, use in research laboratories, and the practical challenges in incorporating these into clinical and diagnostic laboratories, which include high costs, specialised and expensive technologies, difficulties in standardisation, and need for highly trained staff.
- Novel therapies such as CAR-T cells, antibody-drug conjugates, or bispecific antibodies should be included and discussed.
A brief account of these newer biologics-based interventions has been included in section 4.2. and Table 6.
- I like the conclusion, but I suggest adding more actionable recommendations for future research or clinical applications, in addition to the general advocacy for personalized medicine.
Additional actionable recommendations for future research and clinical applications have been included in our conclusion. We suggest a potential need for further screening prior to therapy, necessitating further understanding of the practical limitations of incorporating such investigations into clinical practice, with outsourcing of clinical investigations as a recommended potential solution.